# The Self-Reported Leeds Assessment of Neuropathic Symptoms and Signs (S-LANSS) and PainDETECT Questionnaires in COVID-19 Survivors with Post-COVID Pain

**DOI:** 10.3390/v14071486

**Published:** 2022-07-07

**Authors:** César Fernández-de-las-Peñas, Juan Antonio Valera-Calero, Manuel Herrero-Montes, Pablo del-Valle-Loarte, Rafael Rodríguez-Rosado, Diego Ferrer-Pargada, Lars Arendt-Nielsen, Paula Parás-Bravo

**Affiliations:** 1Department of Physical Therapy, Occupational Therapy, Rehabilitation and Physical Medicine, Universidad Rey Juan Carlos, Alcorcón, 28922 Madrid, Spain; cesar.fernandez@urjc.es; 2Center for Neuroplasticity and Pain (CNAP), SMI, Department of Health Science and Technology, Faculty of Medicine, Aalborg University, DK-9220 Aalborg, Denmark; lan@hst.aau.dk; 3Department of Physiotherapy, Faculty of Health, Universidad Camilo José Cela, 28692 Villanueva de la Cañada, Spain; 4VALTRADOFI Research Group, Department of Physiotherapy, Faculty of Health, Camilo Jose Cela University, 28692 Villanueva de la Cañada, Spain; 5Departamento de Enfermería, Universidad de Cantabria, 39008 Santander, Spain; manuel.herrero@unican.es (M.H.-M.); paula.paras@unican.es (P.P.-B.); 6Grupo de Investigación en Enfermería, Instituto de Investigación Sanitaria Valdecilla (IDIVAL), 39011 Santander, Spain; 7Department of Internal Medicine, Hospital Universitario Severo Ochoa, 28911 Leganes, Spain; pablo.valle@salud.madrid.org (P.d.-V.-L.); Rafael.rodriguez@salud.madrid.org (R.R.-R.); 8Servicio de Neumología, Hospital Universitario Marqués de Valdecilla, 39008 Santander, Spain; diegojose.ferrer@scsalud.es; 9Department of Medical Gastroenterology, Mech-Sense, Aalborg University Hospital, DK-9000 Aalborg, Denmark

**Keywords:** COVID-19, pain, long-COVID, neuropathic, S-LANSS, PainDETECT

## Abstract

This study aimed to analyze correlations between Self-Report Leeds Assessment of Neuropathic Symptoms (S-LANSS) and PainDETECT with proxies of sensitization, pain-related, or psychological/cognitive variables in coronavirus disease, 2019 (COVID-19) survivors exhibiting post-COVID pain. Demographic, clinical, psychological, cognitive, sensitization-associated symptoms, and health-related quality of life were collected in 146 survivors with post-COVID pain. The PainDETECT and S-LANSS questionnaires were used for assessing neuropathic pain-related symptoms. Patients were assessed with a mean of 18.8 (SD 1.8) months after hospitalization. Both questionnaires were positively associated with pain intensity (*p* < 0.05), anxiety (PainDETECT *p* < 0.05; S-LANSS *p* < 0.01), sensitization-associated symptoms (*p* < 0.01), catastrophism (*p* < 0.01), and kinesiophobia (*p* < 0.01) and negatively associated with quality of life (PainDETECT *p* < 0.05; S-LANSS *p* < 0.01). Depressive levels were associated with S-LANSS (*p* < 0.05) but not with PainDETECT. The stepwise regression analyses revealed that 47.2% of S-LANSS was explained by PainDETECT (44.6%), post-COVID pain symptoms duration (1.7%), and weight (1.1%), whereas 51.2% of PainDETECT was explained by S-LANSS (44.6%), sensitization-associated symptoms (5.4%), and anxiety levels (1.2%). A good convergent association between S-LANSS and PainDETECT was found. Additionally, S-LANSS was associated with symptom duration and weight whereas PainDETECT was associated with sensitization-associated symptoms and anxiety levels, suggesting that the two questionnaires evaluate different aspects of the neuropathic pain spectrum in post-COVID pain patients.

## 1. Introduction

There is increasing evidence supporting those individuals who had survived an acute infection of Severe Acute Respiratory Syndrome Coronavirus-2 (SARS-CoV-2), the agent causing coronavirus disease 2019 (COVID-19), developing a plethora of symptoms after the acute phase [1,2]. Among the hundreds of symptoms associated with the post-acute phase, fatigue and dyspnea are the most prevalent post-COVID symptoms [1,2]; however, evidence on post-COVID pain (in particular, headache and musculoskeletal pain) as a prevalent problem is arising [3]. Phenotyping of post-COVID pain could increase current understanding of potential mechanisms and orientate personalized treatment following mechanism-based classifications and be applied in explaining patients’ conditions. To date, it seems that post-COVID pain has a musculoskeletal origin [4]; however, a neuropathic pain origin is also plausible [5]. An increasing number of studies reported the presence of musculoskeletal post-COVID pain in up to 60% of the COVID-19 survivors [6,7,8,9]. The number of studies investigating the prevalence of neuropathic post-COVID pain is lower in relation to those assessing musculoskeletal post-COVID pain [10,11]. These studies have observed that 20% of individuals with post-COVID pain also exhibit neuropathic pain features [10,11].

Musculoskeletal, but also neuropathic, pain can be associated with sensitization-associated symptoms, the underlying mechanisms associated with “nociplastic pain” [12]. Discrimination between musculoskeletal (nociceptive), neuropathic, and nociplastic pain represents a current challenge since one condition does not exclude the other [13]. In fact, all these conditions are not just associated with exaggerated pain responses, but also with central-nervous-system-derived symptoms such as fatigue, sleep problems, memory loss, and psychological disturbances [14]. All these associated symptoms are commonly reported in the plethora of manifestations experienced by individuals with long-COVID [1,2].

Patient-reported outcome measures (PROM) consist of generic or disease-specific self-reported questionnaires assessing different aspects of a condition. Two PROMs used for identifying individuals with neuropathic pain and applied in clinical practice are the Self-Report Leeds Assessment of Neuropathic Symptoms (S-LANSS) [15] and the PainDETECT [16] questionnaires. Both questionnaires have been used in fibromyalgia syndrome [17], a condition which for some patients resemble features similar to long-COVID [18]. In addition, a recent study observed that, although both PROMs are designed for evaluating neuropathic pain symptoms and exhibit good convergence between them, they are related to different pain sensitization proxies [19].

The aims of the current study are: (1) to analyze the differences in identification of neuropathic symptoms between the use of the S-LANSS and PainDETECT in individuals with post-COVID pain; (2) to determine the associations between S-LANSS and PainDETECT with proxies of sensitization, pain-related, or psychological/cognitive variables in people with post-COVID pain; and (3) to conduct a linear regression model to potentially explain the variance and identify those variables contributing to either S-LANSS or PainDETECT to explore possible complementary information provided by both PROMs.

## 2. Materials and Methods

### 2.1. Participants

An observational cross-sectional study was conducted. Patients who had recovered from acute SARS-CoV-2 infection at three urban hospitals in Spain during the first wave in 2020 of the pandemic were screened for eligibility. They were included if had a diagnosis of SARS-CoV-2 infection confirmed by real-time reverse transcription-polymerase chain reaction (RT-PCR) assay of nasopharyngeal and oral swab samples and the presence of consistent clinical/radiological findings at hospitalization and if the presence of “de novo” pain symptoms were reported, starting after the infection for at least three consecutive months. They were excluded if they presented: (1) previous history of pain before the infection; and (2) any other medical comorbidity which could best explain pain, e.g., arthritis. The Local Institutional Ethics Committees (INDIVAL Cantabria 2020.416; HUIL/092-20, HUFA 20/126URJC0907202015920; HSO25112020) approved the study design. Patients were informed of the study and provided their written informed consent prior to their inclusion.

A questionnaire including demographic (gender, age, weight, height), clinical data (intensity and duration of pain symptoms), and several PROMs assessing sensitization-associated symptoms, neuropathic pain, anxiety levels, depressive levels, sleep quality, catastrophism, kinesiophobia, and health-related quality of life was used for data collection.

### 2.2. Neuropathic Pain Assessment

The Self-Report Leeds Assessment of Neuropathic Symptoms (S-LANSS) and the PainDETECT were used for evaluating the presence of the neuropathic pain component.

The S-LANSS uses a binary response where individuals confirm whether they suffer from different symptoms to classify if their pain presents a predominant or non-predominant neuropathic origin. The total score ranges from 0 to 24 points, where ≥12 points suggest the presence of neuropathic pain symptomatology [15]. The S-LANSS has shown proper sensitivity (78%), good internal consistency and validity for identifying neuropathic pain symptoms [15]. This questionnaire is accessible at https://bpac.org.nz/BPJ/2016/May/docs/s-lanss.pdf (acceded on 10 January of 2021).

The PainDETECT is another PROM used to determine the presence of neuropathic pain with a sensitivity of 85% and specificity of 80%. It consists of 9 pain-related items completed in different scales leading to a total score ranging from 0 to 38 points [16]. The PainDETECT uses the following cut-off scores: unlikely neuropathic pain origin (<12 points), ambiguous neuropathic pain origin (12–18 points), or likely neuropathic origin (>18 points) [16]. The questionnaire can be found at https://www.oregon.gov/oha/HPA/dsi-pmc/PainCareToolbox/PainDETECT.pdf (acceded on 10 January of 2021). 

### 2.3. Sensitization-Associated Symptoms

The Central Sensitization Inventory (CSI) evaluates 25 health-related symptoms assumed as a proxy to represent some aspects of sensitization based on a 5-point Likert scale rating [20]. The total score ranges from 0 to 100, where >40 points suggest the presence of sensitization-associated symptoms [21]. The CSI as a single score had good psychometric properties for assessing sensitization-associated symptoms in individuals with chronic pain [22].

### 2.4. Psychological Variables

The Hospital Anxiety and Depression Scale (HADS) was used to evaluate anxiety (HADS-A, 7-items) and depressive (HADS-D, 7-items) levels [23]. The score of each scale ranges from 0 to 21 points, where higher scores suggest more anxiety/depressive levels [23]. Sleep quality was assessed with the Pittsburgh Sleep Quality Index (PSQI) [24]. This PROM consists of 19 self-rated questions (rated on a 4-points Likert scale from 0 to 3) assessing different aspects of sleep (e.g., usual bedtime, wake-up time, number of hours slept, and time needed to fall asleep) during the previous month. The score ranges from 0 to 21 points, where higher scores are suggestive of worse sleep quality [24].

### 2.5. Cognitive Variables

Kinesiophobia, defined as an excessive, irrational, and debilitating fear to perform a physical movement, due to a feeling of vulnerability to a painful injury or reinjury was assessed with the 11-item Tampa Scale Kinesiophobia (TSK-11) [25]. This PROM includes 11 questions where the patient chooses how much they agree or disagree with each (1: “complete disagreement”; 4: “complete agreement”) leading to a total score ranging from 0 to 44 points [25]. Pain catastrophizing, defined as an exaggerated negative mental state brought to bear during an actual or anticipated painful experience, was assessed with the Pain Catastrophizing Scale (PCS) [26]. This PROM consists of 13 items (rated 0: never; 4: always) evaluating rumination, magnification, and despair aspects in relation to the pain experience and leading to a score ranging from 0 to 52 points [26].

### 2.6. Health-Related Quality of Life

Health-related quality of life was evaluated with the paper-based five-level version of EuroQol-5D (EQ-5D-5L) [27]. This general PROM evaluates mobility, self-care, daily activities, pain, and depression/anxiety dimensions into a 5-point Likert scale from 0 (no problems) to 4 (severe problems) points. Responses were converted into a single index number ranging from 0 (equivalent to death) to 1 (optimal health), by applying crosswalk index values for life in Spain [28].

### 2.7. Sample Size Determination

It has been previously suggested that linear regression models require only two subjects per variable for adequate coefficient estimations [29]; however, recent proposals suggest that an adequate sample size for regression models should include 10–15 subjects/variable with no more than 5 variables in the model [30]. Accordingly, for five variables, a minimum of 75 participants is required. To identify the highest number of variables that could be associated with S-LANSS or PainDETECT and for avoiding potential type II errors, we duplicated the estimated sample size.

### 2.8. Statistical Analysis

Descriptive analyses (means and standard deviations, SD) were used to describe the sample. Between-gender differences were assessed with independent Student t-tests. Linear correlations between all variables and both S-LANSS and PainDETECT variables were firstly assessed with Pearson correlation coefficients (r). The bivariate correlation analysis was used to identify multicollinearity and shared variance between the variables (r > 0.8). All statistically significant variables associated with S-LANSS or PainDETECT were included in a stepwise multiple linear regression model (a hierarchical regression analysis) to identify those independent variables contributing significantly to the variance of S-LANSS or PainDETECT, respectively, except variables showing multicollinearity. The significance criterion of the F value for entry into the regression equation was set at *p* < 0.05. Changes in adjusted *R*^2^ were reported after each step of the regression model to determine the association of the additional variables.

## 3. Results

### 3.1. Clinical Data of the Sample

From 200 patients with post-COVID symptoms screened for participation, after verifying they fulfilled all criteria, 146 (73%) participants were included and analyzed. The reason for exclusion of 54 participants was because their main post-COVID symptom was fatigue or dyspnea, but not pain. Participants were assessed at a follow-up period of 18.8 ± 1.8 months after hospital discharge.

Table 1 details clinical, sensory-related, quality of life, and psychological features of the total sample and by gender. The intensity of post-COVID pain, PainDETECT score, sensitization-associated symptoms (CSI), sleep quality (PSQI), and anxiety (HADS-A) and kinesiophobia (TSK-11) levels were significantly higher in females than in males.

Thirty-eight (26%) individuals showed neuropathic pain symptoms according to the S-LANSS (score ≥ 12 points) whereas just 18 (12.2%) patients were classified as likely neuropathic origin (>18 points) and 13 (8.8%) individuals were classified as ambiguous neuropathic pain origin (12–18 points) according to the PainDETECT.

### 3.2. Bivariate Correlation Analysis

Table 2 details bivariate correlation analyses. Multiple correlations for PainDETECT and S-LANSS scores were found. Both questionnaires were positively associated with pain intensity (*p* < 0.05), anxiety (PainDETECT *p* < 0.05, S-LANSS *p* < 0.01), sensitization-associated symptoms (*p* < 0.01), catastrophism (*p* < 0.01), and kinesiophobia levels (*p* < 0.01), and negatively associated with quality of life (PainDETECT *p* < 0.05, S-LANSS *p*< 0.01). Depressive levels were also positively associated with S-LANSS score (*p* < 0.05), but not with PainDETECT. Other associations between the different variables were also identified, but they were not related to the aim of the study.

### 3.3. Multiple Regression Analysis

The hierarchical regression analyses used to determine the explained variance of S-LANSS and PainDETECT scores are summarized in Table 3 and Table 4, respectively. Stepwise regression analyses revealed that PainDETECT score (contributing 44.6%), post-COVID symptoms duration (additional 1.7%), and weight (additional 1.1%) were significantly associated with S-LANSS, explaining 47.4% of its variance (r^2^ adjusted: 0.474, Table 3, Figure 1).

Regarding the PainDETECT regression model, S-LANSS score was an important contributor (explaining 44.6% of the variance), while CSI score and HADS-A explained 5.4% and 1.2% of the PainDETECT score variance, respectively. Therefore, up to 51.2% of S-LANSS variance was explained based on this three-steps model (r^2^ adjusted: 0.512, Table 4, Figure 2).

## 4. Discussion

This is the first study using the PainDETECT and its association with the S-LANSS and other proxies of neurogenic sensitization in COVID-19 survivors with post-COVID pain. The results revealed good convergent association between S-LANSS and PainDETECT, since each questionnaire explained almost 45% of the variance of the other one. Further, the S-LANSS was associated with symptoms duration and weight whereas PainDETECT was associated with sensitization-associated symptoms and anxiety. Our results suggest, as it has been previously observed in women with FMS [19], that these PROMs could assess different aspects of the neuropathic pain spectrum providing synergistic information for phenotyping patients.

The general prevalence of neuropathic symptoms (by using the DN4 questionnaire—Douleur Neuropathique 4 Questions) in individuals with chronic pain has been reported to be 7% [31]. Previous studies on COVID-19 survivors reported prevalence rates of 20% [10,11]. The first study determined the presence of neuropathic symptoms by using a generic question without using a specific PROM [10] whereas the second one used the S-LANSS [11]. In the current study, by using the S-LANSS (cut-off ≥ 12 points) 26% (*n* = 38) of individuals with post-COVID pain exhibited neuropathic symptoms, whereas by using the PainDETECT (cut-off > 18 points) 12.2% (*n* = 18) of the patients exhibited neuropathic likely symptoms. Differences in terminology or particular questions could explain these discrepancies between both PROMs. It is also possible that S-LANSS and PainDETECT assess different components of the neuropathic pain spectrum as previously suggested (19). Studies using objective tests, e.g., quantitative sensory testing, electromyography, or tissue biopsies, confirming the presence of a neuropathic origin of the pain could help to further validate which PROM, the S-LANSS or PainDETECT, leads to a more accurate identification of neuropathic symptoms in individuals with post-COVID pain.

A convergent association between the S-LANSS and PainDETECT seems to be expected since both questionnaires were designed to identify neuropathic pain symptoms [15,16]. This association between both PROMs has been previously found in FMS [19] and painful knee osteoarthritis [32]. However, variables independently associated with each questionnaire were different suggesting that both PROMs could assess different aspects of the neuropathic pain spectrum. The S-LANSS was associated with the duration of post-COVID symptoms and weight. The first association supports that the longer the period with pain, e.g., temporal summation of nociception, the higher the potential development of neuropathic pain symptomatology. This result would suggest that early treatment of pain symptoms in long haulers could be crucial for reducing the risk of neuropathic pain symptomatology. Second, the association between weight (e.g., obesity) and neuropathic pain is also discussed in the literature [33]. Accordingly, this association would explain why exercise programs should be implemented into a multimodal therapeutic approach of long-COVID [34].

The PainDETECT was associated with sensitization-associated symptoms, i.e., the CSI score, and anxiety. The fact that anxiety and stress are related to higher scores on PainDETECT agree with a previous report in people with chronic pain [35]. In our study, the contribution of anxiety levels to PainDETECT in individuals with post-COVID pain was relatively small. In fact, sensitization-associated symptoms had a higher influence on PainDETECT score than anxiety levels. The role of sensitization-associated mechanisms in neuropathic pain is taking particular relevance in the current literature [36]. Interestingly, anxiety could be also considered a sensitization-associated symptom. Indeed, it has been previous suggested that CSI has significant overlap with the psychological construct [37,38,39]. Accordingly, management of stress and sensitization-associated symptoms could also lead to a reduction in symptoms of neuropathic origin in this population.

Potential limitations associated to the current study are recognized. First, we only included previously hospitalized COVID-19 survivors. We do not know the prevalence of neuropathic pain symptomatology in non-hospitalized patients. Second, the prevalence rate of neuropathic symptoms identified in our sample of individuals with post-COVID pain should be based on the use of these specific PROMs. We do not know if the use if other PROMs, e.g., Neuropathic Pain Symptoms Inventory (NPSI) [40] or the DN4 would provide different information. Future longitudinal studies investigating if the associations observed in this study are clinically relevant for the management of long-COVID are now needed.

## 5. Conclusions

The current study observed a convergent association between S-LANSS and PainDETECT, since each questionnaire explained almost 45% of the variance of the other one in previously hospitalized COVID-19 survivors with post-COVID pain. Further, the S-LANSS was associated with symptom duration and weight whereas PainDETECT was associated with sensitization-associated symptoms and anxiety levels, suggesting that both PROMs could assess different aspects of the neuropathic pain spectrum and thereby add synergistic information for phenotyping patients and provide the best possible information to patients about their symptoms. Finally, clinical measurements during the patients’ hospitalization were not collected. Further studies could clarify whether virus proxy measurements (e.g., viral load) or host immune response parameters (e.g., cytokine measurements) can be associated with post-COVID syndrome.

## Figures and Tables

**Figure 1 viruses-14-01486-f001:**
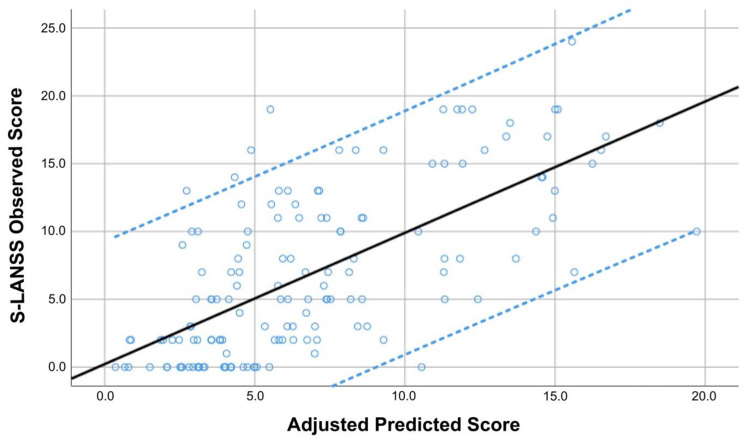
Scatter plot of the adjusted predicted score (r^2^ adjusted: 0.474) explaining the Self-Report Leeds Assessment of Neuropathic Symptoms (S-LANSS) in COVID-19 survivors with post-COVID pain symptoms (*n* = 146). Note that some points can be overlapping. Black line represents de mean predicted score whereas the blue lines represent the 95% confidence intervals.

**Figure 2 viruses-14-01486-f002:**
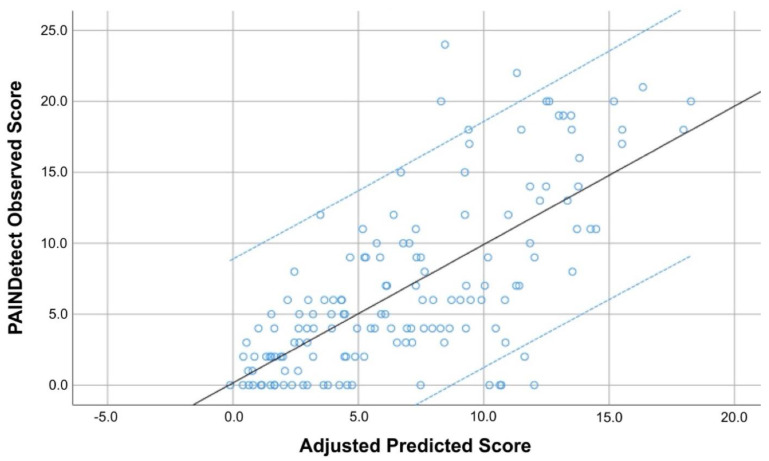
Scatter plot of the adjusted predicted score (r^2^ adjusted: 0.512) explaining the PainDETECT score in COVID-19 survivors with post-COVID pain symptoms (*n* = 146). Note that some points can be overlapping. Black line represents de mean predicted score whereas the blue lines represent the 95% confidence intervals.

**Table 1 viruses-14-01486-t001:** Baseline outcomes (mean ± SD) of the sample.

Baseline Variable	Sample (*n* = 146)	Males (*n* = 67)	Females (*n* = 78)
Demographic Characteristics			
Age (years)	57.5 ± 11.8	60.0 ± 10.3	55.2 ± 12.5
Height (m)	1.67 ± 0.09	1.73 ± 0.08	1.61 ± 0.06
Weight (kg)	81.8 ± 17.1	86.5 ± 15.6	77.8 ± 17.4
*Clinical Characteristics*			
Pain intensity (NPRS, 0–10)	5.6 ± 1.7	5.2 ± 1.85	5.9 ± 1.55
Post-COVID symptoms (months)	18.8 ± 1.8	18.7 ± 2.0	18.9 ± 1.7
Sensory-Related Features			
PainDETECT (0–38)	6.8 ± 6.1	5.5 ± 5.5	7.9 ± 6.45
S-LANSS (0–24)	6.9 ± 6.1	6.45 ± 5.9	7.2 ± 6.25
Central Sensitization Inventory (0–100)	33.9 ± 17.25	25.9 ± 14.3	41.05 ± 16.45
Quality of Life			
EuroQol 5-D Questionnaire (0–1)	0.75 ± 0.2	0.8 ± 0.2	0.75 ± 0.2
Psychological Characteristics			
HADS-A (0–21)	5.3 ± 4.2	4.45 ± 4.05	6.1 ± 4.2
HADS-D (0–21)	5.1 ± 4.3	4.4 ± 4.3	5.6 ± 4.3
Pittsburgh Sleeping Quality Index (0–21)	8.1 ± 4.3	6.85 ± 4.4	9.1 ± 3.9
Pain Catastrophizing Scale (0–52)	12.15 ± 11.95	10.3 ± 11.3	13.8 ± 12.4
Tampa Scale for Kinesiophobia (0–44)	24.1 ± 8.55	22.5 ± 8.75	25.5 ± 8.25

NPRS: Numerical Pain Rate Scale; HADS: Hospital Anxiety and Depression Scale (A: Anxiety; D: Depression); S-LANSS: self-reported version of the Leeds Assessment of Neuropathic Symptoms and Signs.

**Table 2 viruses-14-01486-t002:** Pearson-product moment correlation matrix between sociodemographic, psychological, neuro-physiological and clinical characteristics.

	1	2	3	4	5	6	7	8	9	10	11	12	13	14
1. Age														
2. Gender	−0.206 *													
3. Height	n.s.	−0.595 **												
4. Weight	n.s.	−0.256 **	0.509 **											
5. Post-COVID Symptoms	n.s.	n.s.	n.s.	n.s.										
6. Pain intensity	n.s.	0.200 *	−0.191 *	n.s.	n.s.									
7. HADS-A	n.s.	0.194 *	n.s.	n.s.	−0.271 **	0.175 *								
8. HADS-D	n.s.	n.s.	n.s.	n.s.	n.s.	0.225 **	0.750 **							
9. PSQI	n.s.	0.262 **	−0.213 **	n.s.	−0.189 *	n.s.	0.316 **	0.354 **						
10. PAINDetect	n.s.	0.193 *	−0.254 **	n.s.	n.s.	0.198 *	0.169 *	n.s.	n.s.					
11. S-LANSS	n.s.	n.s.	n.s.	0.193 *	n.s.	0.212 *	0.213 **	0.169 *	n.s.	0.671 **				
12. CSI	n.s.	0.440 **	−0.285 **	n.s.	n.s.	0.190 *	0.551 **	0.446 **	0.390 **	0.413 **	0.274 **			
13. PCS	n.s.	n.s.	n.s.	n.s.	−0.343 **	n.s.	0.492 **	0.483 **	0.282 **	0.220 **	0.263 **	0.402 **		
14. TSK-11	n.s.	0.168 *	n.s.	n.s.	n.s.	n.s.	0.356 **	0.306 **	0.288 **	0.283 *	0.303 **	0.450 **	0.578 **	
15. EuroQol 5-D	n.s.	n.s.	n.s.	n.s.	n.s.	n.s.	n.s.	−0.174 *	−0.301 **	n.s.	n.s.	−0.199 *	−0.210 *	n.s.

HADS: Hospital Anxiety and Depression Scale (A: Anxiety; D: Depression); S-LANSS: self-reported version of the Leeds Assessment of Neuropathic Symptoms and Signs; CSI: Central Sensitization Inventory. * *p* < 0.05; ** *p* < 0.01.

**Table 3 viruses-14-01486-t003:** Stepwise regression analyses to determine predictors of self-reported version of the Leeds Assessment of Neuropathic Symptoms and Signs (S-LANSS) score.

	Predictor Outcome	Β	SE B	95% CI	*B*	t	*p*
S-LANSS	Step 1 PainDETECT	0.670	0.062	0.548; 0.792	0.671	10.860	<0.001
	Step 2 PainDETECT Post-COVID Duration	0.666 −0.467	0.061 0.203	0.545; 0.786 −0.867; −0.066	0.667 −0.140	10.950 −2.305	<0.001 0.023
	Step 3 PainDETECT Post-COVID Duration Weight	0.667 −0.518 0.044	0.060 0.202 0.022	0.548; 0.786 −0.918; −0.119 0.001; 0.087	0.669 −0.156 0.122	11.092 −2.564 2.006	<0.001 0.011 0.047

r^2^ adj. = 0.446 for step 1, r^2^ adj. = 0.463 for step 2, r^2^ adj. = 0.474 for step 3.

**Table 4 viruses-14-01486-t004:** Summary of the stepwise regression analyses to determine predictors of PainDETECT.

	Predictor Outcome	Β	SE B	95% CI	*B*	t	*p*
PainDETECT	Step 1 S-LANSS	0.672	0.062	0.550; 0.795	0.671	10.860	<0.001
	Step 2 S-LANSS CSI	0.604 0.088	0.061 0.022	0.483; 0.725 0.045; 0.130	0.603 0.247	9.876 4.042	<0.001 <0.001
	Step 3 S-LANSS CSI HADS-A	0.614 0.116 –0.215	0.061 0.025 0.101	0.494; 0.734 0.066; 0.165 −0.415; −0.015	0.613 0.326 −0.148	10.129 4.596 −2.120	<0.001 <0.001 0.036

CSI: Central Sensitization Inventory; S-LANSS: self-reported version of the Leeds Assessment of Neuropathic Symptoms and Signs; HADS: Hospital Anxiety and Depression Scale (A: Anxiety). R^2^ adj. = 0.446 for step 1, R^2^ adj. = 0.500 for step 2, R^2^ adj. = 0.512 for step 3.

## Data Availability

All data relevant to the study are included in the article.

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
