# Peer review of "The Self-Reported Leeds Assessment of Neuropathic Symptoms and Signs (S-LANSS) and PainDETECT Questionnaires in COVID-19 Survivors with Post-COVID Pain"

_viruses, 2022, doi:10.3390/v14071486_

Round 1
Reviewer 1 Report
The study “The Self-Reported Leeds Assessment of Neuropathic Symptoms and Signs (S-LANSS) and PainDETECT Questionnaires in COVID-19 Survivors with post-COVID Pain” by the authors is a well conducted study. The authors assessed well used patient-reported outcome measures (PROM) to identify possible neuropathic pain in post-COVID pain patients. Results are well analyzed and presented. This can be an important first-step in patient care. While the article has been submitted to “Post-COVID Syndrome” special issue, I am not sure about the appropriateness of the article for the journal “Viruses”. For e.g.., most of the cited literature understandably is from other specialized journals.
However, to better align the article with the broader audience of Viruses, I recommend the following suggestions:
- The authors must provide links or as supplemental data all the questionnaires that were used in this study. This will help other researchers carry out similar studies.
- The authors must provide clinical measurements made during the hospital stay for these COVID-19 patients and perform correlation/regression analysis to identify (if any) virus proxy measurements (e.g., viral load) or host immune response parameters (e.g., cytokine measurements) that can be associated with post-COVID syndrome.
Author Response
Response Letter manuscript viruses-1782108
The Self-Reported Leeds Assessment of Neuropathic Symptoms and Signs (S-LANSS) and PainDETECT Questionnaires in COVID-19 Survivors with post-COVID Pain
We would like to thank the reviewers for their comments, which we believe have clarified many aspects of the manuscript. We have edited the text according to the suggestions from the reviewers. We have highlighted all changes in light grey throughout the manuscript. A point-by-point response is presented below.
Reviewer 1
The study “The Self-Reported Leeds Assessment of Neuropathic Symptoms and Signs (S-LANSS) and PainDETECT Questionnaires in COVID-19 Survivors with post-COVID Pain” by the authors is a well conducted study. The authors assessed well used patient-reported outcome measures (PROM) to identify possible neuropathic pain in post-COVID pain patients. Results are well analyzed and presented. This can be an important first-step in patient care. While the article has been submitted to “Post-COVID Syndrome” special issue, I am not sure about the appropriateness of the article for the journal “Viruses”. For e.g.., most of the cited literature understandably is from other specialized journals. However, to better align the article with the broader audience of Viruses, I recommend the following suggestions:
Response: Thank you for the positive feedback.
The authors must provide links or as supplemental data all the questionnaires that were used in this study. This will help other researchers carry out similar studies.
Response: A link for both questionnaires has been provided.
The authors must provide clinical measurements made during the hospital stay for these COVID-19 patients and perform correlation/regression analysis to identify (if any) virus proxy measurements (e.g., viral load) or host immune response parameters (e.g., cytokine measurements) that can be associated with post-COVID syndrome.
Response: We regret to comment that clinical data from the hospitals were not accessible for the research team. We added this point in limitations.
We would like to thank the reviewers and we hope that the current version of the paper can be accepted in Viruses
Sincerely yours,
The authors
Reviewer 2 Report
This manuscript studies and analyzes correlation between two FROM questionnaires: S-LANSS and PainDETECT in COVID-19 convalescent individuals exhibiting post-COVID pain. The results reveal great association between two questionnaires while they would possibly access different aspects of the neuropathic pain spectrum. The method, results, and discussion in this manuscript are clearly explained. A few minor suggestions: the definition of S-LANSS in line 79 can be deleted since it has been defined above; change "weigh" to "weight" in lines 258 and 321.
Author Response
Response Letter manuscript viruses-1782108
The Self-Reported Leeds Assessment of Neuropathic Symptoms and Signs (S-LANSS) and PainDETECT Questionnaires in COVID-19 Survivors with post-COVID Pain
We would like to thank the reviewers for their comments, which we believe have clarified many aspects of the manuscript. We have edited the text according to the suggestions from the reviewers. We have highlighted all changes in light grey throughout the manuscript. A point-by-point response is presented below.
Reviewer 2
This manuscript studies and analyzes correlation between two FROM questionnaires: S-LANSS and PainDETECT in COVID-19 convalescent individuals exhibiting post-COVID pain. The results reveal great association between two questionnaires while they would possibly access different aspects of the neuropathic pain spectrum. The method, results, and discussion in this manuscript are clearly explained. A few minor suggestions: the definition of S-LANSS in line 79 can be deleted since it has been defined above; change "weigh" to "weight" in lines 258 and 321.
Response: Thank you for your kind comments. We modified these two misspellings.
We would like to thank the reviewers and we hope that the current version of the paper can be accepted in Viruses
Sincerely yours,
The authors